# MAXIMUM LIKELIHOOD ESTIMATION FOR MULTI-MODAL LEARNING WITH MISSING MODALITY

## ABSTRACT

Multimodal learning has achieved great successes in many scenarios. Compared with unimodal learning, it can effectively combine the information from different modalities to improve the performance of learning tasks. In reality, the multimodal data may have missing modalities due to various reasons, such as sensor failure and data transmission error. In previous works, the information of the modality-missing data has not been well exploited. To address this problem, we propose an efficient approach based on maximum likelihood estimation to incorporate the knowledge in the modality-missing data. Specifically, we design a likelihood function to characterize the conditional distributions of the modality-complete data and the modality-missing data, which is theoretically optimal. Moreover, we develop a generalized form of the softmax function to effectively implement maximum likelihood estimation in an end-to-end manner. Such training strategy guarantees the computability of our algorithm capably. Finally, we conduct a series of experiments on real-world multimodal datasets. Our results demonstrate the effectiveness of the proposed approach, even when 95% of the training data has missing modality.

## 1 INTRODUCTION

Multimodal learning is an important research area, which builds models to process and relate information between different modalities (Ngiam et al., 2011; Srivastava & Salakhutdinov, 2014; Baltrušaitis et al., 2018). **Compared with unimodal learning, multimodal learning can achieve better performance by properly utilizing the multimodal data.** It has been successfully used in many applications, such as multimodal emotion recognition (Soleymani et al., 2011; Mittal et al., 2020), multimedia event detection (Li et al., 2020), and visual question-answering (Yu et al., 2019). With the emergence of big data, multimodal learning becomes more and more important to combine the multimodal data from different sources.

A number of previous works (Tzirakis et al., 2017; Zhang et al., 2017; Elliott et al., 2017; Kim et al., 2020; Zhang et al., 2020) have achieved great successes based on complete observations during the training process. However, in practice, the multimodal data may have missing modalities (Du et al., 2018; Ma et al., 2021a;b). This may be caused by various reasons. For instance, the sensor that collects the multimodal data is damaged or the network transmission fails. Examples of the multimodal data are shown in Figure 1.

In the past years, different approaches have been proposed to deal with modality missing. A simple and typical way (Hastie et al., 2009) is to directly discard the data with missing modalities. Since the information contained in the modality-missing data is neglected, such method often has limited performance. In addition, researchers (Tran et al., 2017; Chen & Zhang, 2020; Liu et al., 2021; Ma et al., 2021b) have proposed approaches to heuristically combine the information of the modality-missing data. However, most of these works lack theoretical explanations, and these empirical methods are often implemented using multiple training stages rather than an end-to-end manner, which lead to the information of the modality-missing data not being well exploited.

To tackle above issues, we propose an efficient approach based on maximum likelihood estimation to effectively utilize the modality-missing data. To be specific, we present a likelihood function to characterize the conditional distributions of the modality-complete data and the modality-missing data, which is theoretically optimal. Furthermore, we adopt a generalized form of the softmax

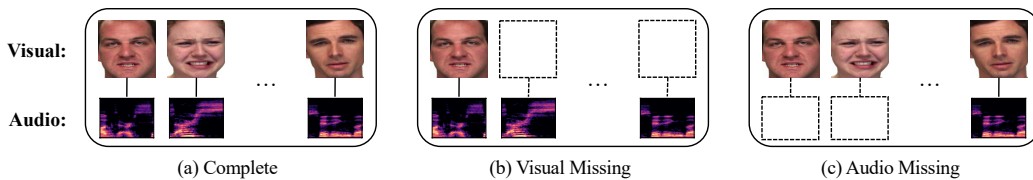

Visual:

Audio:

(a) Complete        (b) Visual Missing        (c) Audio Missing

Figure 1: Examples of the multimodal data: (a) complete observations, (b) observations which may have missing visual modality, and (c) observations which may have missing audio modality.

function to efficiently implement our maximum likelihood estimation algorithm. Such training strategy guarantees the computability of our framework in an end-to-end scheme. In this way, our approach can effectively leverage the information of the modality-missing data during the training process, Finally, we perform several experiments on real-world multimodal datasets, including eNTERFACE'05 (Martin et al., 2006) and RAVDESS (Livingstone & Russo, 2018). The results show the effectiveness of our approach in handling the problem of modality missing. To summarize, our contribution is three-fold:

- We design a likelihood function to learn the conditional distributions of the modality-complete data and the modality-missing data, which is theoretically optimal.

- We develop a generalized form of the softmax function to implement our maximum likelihood estimation framework in an end-to-end manner, which is more effective than previous works.

- We conduct a series of experiments on real-world multimodal datasets. The results validate the effectiveness of our approach, even when 95% of the training data has missing modality.

## 2 METHODOLOGY

Our goal is to deal with the problem of modality missing in multimodal learning based on maximum likelihood estimation. In the following, we first show the problem formulation, and then describe the details of our framework.

### 2.1 PROBLEM FORMULATION

In this paper, we consider that the multimodal data has two modalities. Here, the random variables corresponding to these two modalities and their category labels are denoted as $X$, $Y$, and $Z$, respectively. In the training process, we assume that there are two independently observed datasets: modality-complete and modality-missing. We use $D_{XYZ} = \left\{ (x_c^{(i)}, y_c^{(i)}, z_c^{(i)}) \mid z_c^{(i)} \in \mathcal{Z} = \{1, 2, \cdots, |\mathcal{Z}|\} \right\}_{i=1}^{n_c}$ to represent the modality-complete dataset, where $x_c^{(i)}$ and $y_c^{(i)}$ represent the two modalities of the $i$-th sample of $D_{XYZ}$ respectively, $z_c^{(i)}$ is their corresponding category label, and the size of $D_{XYZ}$ is $n_c$. We then use $D_{XZ} = \left\{ (x_m^{(i)}, z_m^{(i)}) \mid z_m^{(i)} \in \mathcal{Z} = \{1, 2, \cdots, |\mathcal{Z}|\} \right\}_{i=1}^{n_m}$ to represent the modality-missing dataset, where the size of $D_{XZ}$ is $n_m$. In addition, we adopt $[D_{XYZ}]_{XY}$ to represent $\left\{ (x_c^{(i)}, y_c^{(i)}) \right\}_{i=1}^{n_c}$. $[D_{XYZ}]_Z$, $[D_{XZ}]_X$, and $[D_{XZ}]_Z$ are expressed in the same way. The multimodal data of $D_{XYZ}$ and $D_{XZ}$ are assumed to be i.i.d. generated from an unknown underlying joint distribution. By utilizing the knowledge of the modality-complete data and the modality-missing data, we hope our framework can predict the category labels correctly.

### 2.2 MAXIMUM LIKELIHOOD ESTIMATION FOR MISSING MODALITY

In this section, we first present how to design a likelihood function to learn the conditional distributions of the modality-complete data and the modality-missing data. Then, we show that by adopting a generalized form of the softmax function, we design a training strategy to implement our algorithm.

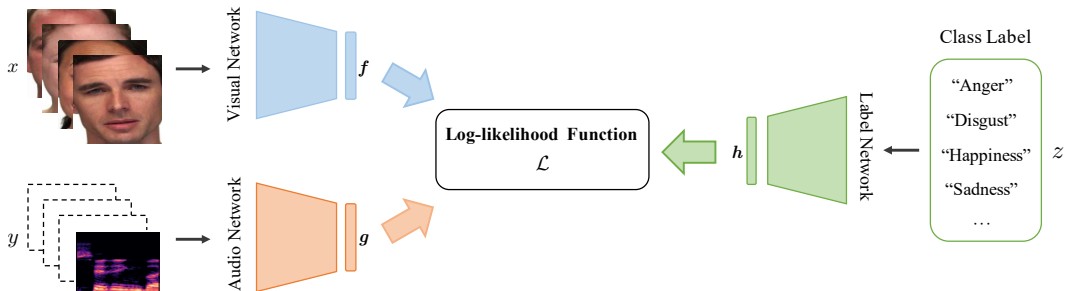

Figure 2: Our proposed framework for multimodal learning with missing modality. In the training process, we propose a log-likelihood function $\mathcal{L}$, as shown in Equation (2), to learn the conditional distributions of the modality-complete data and the modality-missing data. By developing a generalized form of the softmax function, we implement our maximum likelihood estimation algorithm in an end-to-end manner.

### 2.2.1 LIKELIHOOD FUNCTION ANALYSES

Maximum likelihood estimation is a statistical method of using the observed data to estimate the distribution by maximizing the likelihood function. The estimated distribution makes the observed data most likely (Myung, 2003). With this idea, we study the likelihood function on datasets $D_{XYZ}$ and $D_{XZ}$. For the classification task, the conditional likelihood is commonly used. Inspired by this, we use a model $Q_{XYZ}$ to learn the underlying joint distribution of $D_{XYZ}$ and $D_{XZ}$. The conditional likelihood can be represented as:

$$
\begin{aligned}
\ell &\triangleq \mathbb{P}\left([D_{XYZ}]_Z, [D_{XZ}]_Z \mid [D_{XYZ}]_{XY}, [D_{XZ}]_X; Q_{XYZ}\right) \\
&\overset{\mathbf{a}}{=} \mathbb{P}\left([D_{XYZ}]_Z \mid [D_{XYZ}]_{XY}; Q_{XYZ}\right) \cdot \mathbb{P}\left([D_{XZ}]_Z \mid [D_{XZ}]_X; Q_{XYZ}\right) \\
&\overset{\mathbf{b}}{=} \prod_{(x,y,z)\in D_{XYZ}} Q_{Z|XY}(z|xy) \cdot \prod_{(x,z)\in D_{XZ}} Q_{Z|X}(z|x)
\end{aligned}
\tag{1}
$$

where the step $\mathbf{a}$ follows from the fact that datasets $D_{XYZ}$ and $D_{XZ}$ are observed independently, and the step $\mathbf{b}$ is due to that samples in each dataset are i.i.d. $Q_{Z|XY}$ and $Q_{Z|X}$ are conditional distributions of $Q_{XYZ}$. In this way, we show the likelihood function using the information of $D_{XYZ}$ and $D_{XZ}$. Then, we use the negative log-likelihood as the loss function to train our deep learning model, i.e.,

$$
\mathcal{L} \triangleq -\log \ell = -\sum_{(x,y,z)\in D_{XYZ}} \log Q_{Z|XY}(z|xy) - \sum_{(x,z)\in D_{XZ}} \log Q_{Z|X}(z|x)
\tag{2}
$$

It is worth noting that in (Daniels, 1961; Lehmann, 2004), maximum likelihood estimation is proved to be an asymptotically-efficient strategy, which guarantees the theoretical optimality of our method to deal with modality missing.

To optimize $\mathcal{L}$, we use deep neural networks to extract the $k$-dimensional feature representations from the observation $(x, y, z)$, which are represented as $\boldsymbol{f}(x) = [f_1(x), f_2(x), \cdots, f_k(x)]^{\mathrm{T}}$, $\boldsymbol{g}(y) = [g_1(y), g_2(y), \cdots, g_k(y)]^{\mathrm{T}}$, and $\boldsymbol{h}(z) = [h_1(z), h_2(z), \cdots, h_k(z)]^{\mathrm{T}}$, respectively. We then utilize these features to learn $Q_{Z|XY}$ and $Q_{Z|X}$ in $\mathcal{L}$. Our framework is shown in Figure 2.

In this way, we show the log-likelihood function $\mathcal{L}$. By characterizing the conditional distributions of the modality-complete data and the modality-missing data, it leverages the underlying structure information behind the multimodal data, which constitutes the theoretical basis of our framework.

### 2.2.2 MAXIMUM LIKELIHOOD ESTIMATION IMPLEMENTATION

In fact, it is not easy to optimize the log-likelihood function $\mathcal{L}$ in Equation (2) by designing neural networks, which is mainly due to two reasons. Firstly, the representations of the high-dimensional data and the procedure to model them are complicated. Secondly, since $Q_{Z|XY}$ and $Q_{Z|X}$ in $\mathcal{L}$ are

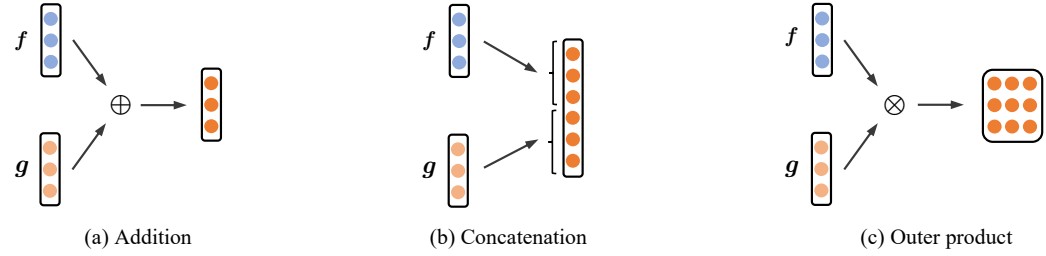

(a) Addition                     (b) Concatenation                 (c) Outer product

Figure 3: Three forms of $\phi$ are studied: (a) addition (Wang et al., 2019), i.e., $\phi(\boldsymbol{f}, \boldsymbol{g}) \triangleq \boldsymbol{f} + \boldsymbol{g}$, (b) concatenation (Chandar et al., 2016), i.e., $\phi(\boldsymbol{f}, \boldsymbol{g}) \triangleq [\boldsymbol{f}^{\mathrm{T}}, \boldsymbol{g}^{\mathrm{T}}]^{\mathrm{T}}$, and (c) outer product (Zadeh et al., 2017), i.e., $\phi(\boldsymbol{f}, \boldsymbol{g}) \triangleq \mathbf{vec}(\boldsymbol{f} \otimes \boldsymbol{g})$, where $\mathbf{vec}$ represents the vectorization of outer product.

related, how to build models to learn their relationships is difficult. To address these two issues, we develop a generalized form of the softmax function to describe $Q_{XYZ}$ as follows [1]:

$$Q_{XYZ}(x, y, z) = \frac{R_X(x)R_Y(y)R_Z(z) \exp(\phi^{\mathrm{T}}(\boldsymbol{f}(x), \boldsymbol{g}(y))\boldsymbol{h}(z))}{\sum_{x',y',z'} R_X(x')R_Y(y')R_Z(z') \exp(\phi^{\mathrm{T}}(\boldsymbol{f}(x'), \boldsymbol{g}(y'))\boldsymbol{h}(z'))} \quad (3)$$

where $\phi(\boldsymbol{f}, \boldsymbol{g})$ represents the function to fuse features $\boldsymbol{f}$ and $\boldsymbol{g}$. We study three forms of $\phi$ to investigate its effect in our framework, as shown in Figure 3. $R_X$, $R_Y$, and $R_Z$ represent the underlying marginal distributions of the variables $X$, $Y$, and $Z$, respectively. Their use makes the denominator of Equation (3) expressed in the form of the mean over $R_X$, $R_Y$, and $R_Z$, which serves as the normalization to make $Q_{XYZ}$ a valid distribution and is helpful for our further derivation. In addition, the generalized softmax function we propose can be regarded as a generalization of softmax learning in (Xu et al., 2018) from unimodal learning to multimodal learning.

In this way, we show the distribution $Q_{XYZ}$ by adopting a generalized form of the softmax function, which has the following two benefits. Firstly, by depicting the representation of $Q_{XYZ}$, we can further derive $Q_{Z|XY}$ and $Q_{Z|X}$. It makes our approach a unified framework to combine the information of the modality-complete data and the modality-missing data. Secondly, it avoids modeling the relationship between $Q_{Z|XY}$ and $Q_{Z|X}$. In fact, the correlation between the high-dimensional data can be rather complex.

Then, we derive conditional distributions $Q_{Z|XY}$ and $Q_{Z|X}$ from Equation (3):

$$Q_{Z|XY}(z|xy) = R_Z(z)\frac{\exp(\phi^{\mathrm{T}}(\boldsymbol{f}(x), \boldsymbol{g}(y))\boldsymbol{h}(z))}{\sum_{z'} R_Z(z') \exp(\phi^{\mathrm{T}}(\boldsymbol{f}(x), \boldsymbol{g}(y))\boldsymbol{h}(z'))} \quad (4)$$

and

$$Q_{Z|X}(z|x) = R_Z(z)\frac{\sum_{y'} R_Y(y') \exp(\phi^{\mathrm{T}}(\boldsymbol{f}(x), \boldsymbol{g}(y'))\boldsymbol{h}(z))}{\sum_{z'} R_Z(z') \sum_{y'} R_Y(y') \exp(\phi^{\mathrm{T}}(\boldsymbol{f}(x), \boldsymbol{g}(y'))\boldsymbol{h}(z'))} \quad (5)$$

We can observe that by introducing $R_X$, $R_Y$, and $R_Z$ into $Q_{XYZ}$, the derived $Q_{Z|XY}$ and $Q_{Z|X}$ are expressed in the form of the mean over $R_Y$ and $R_Z$. In practice, we can use the empirical mean as an estimation. Correspondingly, by plugging Equations (4) and (5) into Equation (2), we can summarize the detailed steps to compute our objective function $\mathcal{L}$, as shown in Algorithm 1. It is worth pointing out that when we compute $Q_{Z|X}$, we need to use the information of the modality $y$. Since in the training process, the modality $y$ of the dataset $D_{XZ}$ is missing, we utilize samples of the modality $y$ of the dataset $D_{XYZ}$ to compute $Q_{Z|X}$.

Finally, we utilize neural networks to extract features $\boldsymbol{f}$, $\boldsymbol{g}$, and $\boldsymbol{h}$ from the modality-complete data and the modality-missing data to optimize our log-likelihood function $\mathcal{L}$. It performs classification directly, which does not need to explicitly complement the modality-missing data before the classification task.

---

[1]Strictly speaking, $R_X$ and $R_Y$ are probability density functions, and $R_Z$ is a probability mass function. The denominator of Equation (3) should be integrated over $R_X$ and $R_Y$. We use summation here for the simplicity of exposition.

---

**Algorithm 1** Compute our objective function on a mini-batch.

---

**Input:**
  A modality-complete batch $\left\{(x_{\mathrm{c}}^{(i)}, y_{\mathrm{c}}^{(i)}, z_{\mathrm{c}}^{(i)})\right\}_{i=1}^{n_1}$, where $n_1$ is the batch size.

  A modality-missing batch $\left\{(x_{\mathrm{m}}^{(i)}, z_{\mathrm{m}}^{(i)})\right\}_{i=1}^{n_2}$, where $n_2$ is the batch size.
  Neural networks with $k$ output units: $\boldsymbol{f}$, $\boldsymbol{g}$, and $\boldsymbol{h}$.
**Output:**
  The value of our objective $\mathcal{L}$.
1: Compute empirical label distribution $\hat{R}_Z$:
  $\hat{R}_Z(z) \leftarrow \frac{\sum_{i=1}^{n_1} \mathbb{1}(z_{\mathrm{c}}^{(i)}=z) + \sum_{i=1}^{n_2} \mathbb{1}(z_{\mathrm{m}}^{(i)}=z)}{n_1 + n_2}, z = 1, 2, \cdots, |\mathcal{Z}|$
2: Compute $Q_{Z|XY}$:
  $Q_{Z|XY}(z_{\mathrm{c}}^{(i)}|x_{\mathrm{c}}^{(i)}, y_{\mathrm{c}}^{(i)}) \leftarrow \hat{R}_Z(z_{\mathrm{c}}^{(i)}) \frac{\exp(\boldsymbol{\phi}^{\mathrm{T}}(\boldsymbol{f}(x_{\mathrm{c}}^{(i)}), \boldsymbol{g}(y_{\mathrm{c}}^{(i)}))\boldsymbol{h}(z))}{\sum_{z'=1}^{|\mathcal{Z}|} R_Z(z') \exp(\boldsymbol{\phi}^{\mathrm{T}}(\boldsymbol{f}(x_{\mathrm{c}}^{(i)}), \boldsymbol{g}(y_{\mathrm{c}}^{(i)}))\boldsymbol{h}(z'))}, \ i = 1, \cdots, n_1$
3: Compute $Q_{Z|X}$:
  $Q_{Z|X}(z_{\mathrm{m}}^{(i)}|x_{\mathrm{m}}^{(i)}) \leftarrow \hat{R}_Z(z_{\mathrm{m}}^{(i)}) \frac{\frac{1}{n_1}\sum_{j=1}^{n_1} \exp(\boldsymbol{\phi}^{\mathrm{T}}(\boldsymbol{f}(x_{\mathrm{m}}^{(i)}), \boldsymbol{g}(y_{\mathrm{c}}^{(j)}))\boldsymbol{h}(z_{\mathrm{m}}^{(i)}))}{\sum_{z'=1}^{|\mathcal{Z}|} R_Z(z') \frac{1}{n_1}\sum_{j=1}^{n_1} \exp(\boldsymbol{\phi}^{\mathrm{T}}(\boldsymbol{f}(x_{\mathrm{m}}^{(i)}), \boldsymbol{g}(y_{\mathrm{c}}^{(j)}))\boldsymbol{h}(z'))}, \ i = 1, \cdots, n_2$

4: Compute our empirical objective $\mathcal{L}$:
  $-\sum_{i=1}^{n_1} \log Q_{Z|XY}(z_{\mathrm{c}}^{(i)}|x_{\mathrm{c}}^{(i)}, y_{\mathrm{c}}^{(i)}) - \sum_{i=1}^{n_2} \log Q_{Z|X}(z_{\mathrm{m}}^{(i)}|x_{\mathrm{m}}^{(i)})$

---

## 3 EXPERIMENTS

In this section, we first describe the real-world multimodal datasets used in our experiment, then explain the experimental settings and baseline methods, and finally give the experimental results to show the effectiveness of our approach.

### 3.1 DATASETS

We perform experiments on two public real-world multimodal datasets: eNTERFACE'05 (Martin et al., 2006) and RAVDESS (Livingstone & Russo, 2018). eNTERFACE'05 is an audio-visual emotion database in English. It contains 42 subjects eliciting the six basic emotions: anger, disgust, fear, happiness, sadness, and surprise. There are 213 videos for happiness, and 216 videos for each of the remaining emotions. Following (Ma et al., 2020), we extract 30 segment samples from each video and then obtain a processed dataset with 38,790 samples.

RAVDESS is a multimodal database of emotional speech and song, which consists of 24 professional actors in a neutral North American accent. Here, we use the speech part, which includes calm, happy, sad, angry, fearful, surprise, and disgust expressions. Each recording is also in the video form. Similar to the eNTERFACE'05 dataset, we only consider six basic emotions, each of which has 5,760 segment samples.

### 3.2 EXPERIMENTAL SETTINGS

We perform experiments on the processed eNTERFACE'05 and RAVDESS datasets. Each segment of these two datasets has a duration of 0.5 seconds. As shown in ((Ma et al., 2020)), consecutive frames within 0.5 seconds usually contain the same emotion in a similar way, which inspired us to choose the central frame of each segment as the visual modality. This technique not only makes that the visual data contains enough emotional information, but also avoids the redundancy in multiple frames. Besides, the log Mel-spectrogram is extracted from each segment as the audio modality, which is similar to the RGB image. We then feed these data into our framework to obtain the classification result. ResNet-50 (He et al., 2016) is used as the backbone of visual network $\mathbf{f}$ and audio network $\mathbf{g}$ to extract features from visual and audio modalities, respectively. In addition, we transform the corresponding label into the one-hot form and then extract the label feature using label network $\mathbf{h}$ with a fully connected layer. $\mathbf{f}$, $\mathbf{g}$, and $\mathbf{h}$ are trained together.

On each processed dataset, we split all data into three parts: training set, validation set, and test set. Their proportions are 70%, 15%, and 15%. In practice, modality missing often occurs with a high

Table 1: The classification performance with missing modality on the eNTERFACE'05 dataset.

| Method | | Visual Missing | | | Audio Missing | | |
|---|---|---|---|---|---|---|---|
| | | 95% | 90% | 80% | 95% | 90% | 80% |
| Lower Bound | Addition | 26.39 | 35.26 | 46.91 | 27.53 | 35.26 | 50.93 |
| | Concatenation | 27.11 | 36.39 | 46.49 | 27.84 | 33.71 | 46.29 |
| | Outer product | 26.91 | 37.53 | 42.78 | 28.56 | 34.95 | 48.14 |
| AE | Addition | 41.24 | 47.11 | 50.31 | 42.78 | 45.77 | 51.75 |
| | Concatenation | 43.92 | 46.39 | 50.00 | 42.16 | 47.42 | 50.72 |
| | Outer product | 49.79 | 52.16 | 55.15 | 47.73 | 53.09 | 53.51 |
| HGR MC | Addition | 41.34 | 59.69 | 58.97 | 54.95 | 74.12 | 77.32 |
| | Concatenation | 41.34 | 57.84 | 63.51 | 57.42 | 75.67 | 79.18 |
| | Outer product | 49.90 | 59.69 | 64.64 | 55.46 | 76.29 | 77.94 |
| ZP | Addition | 58.66 | 67.84 | 69.07 | 76.49 | 78.35 | 80.41 |
| | Concatenation | 60.21 | 67.11 | 68.76 | 76.70 | 78.25 | 80.93 |
| Ours | Addition | **66.29** | **71.65** | **72.37** | 79.38 | 80.31 | 81.24 |
| | Concatenation | 64.74 | 70.82 | 72.27 | 79.79 | 80.21 | 81.24 |
| | Outer product | 66.08 | 71.13 | 72.06 | **80.31** | **81.03** | **81.65** |

missing rate (Suo et al., 2019; Ma et al., 2021b). Here, in the training stage, we study three kinds of missing rates: 80%, 90%, and 95%. The case where the audio modality is missing and the case where the visual modality is missing are investigated respectively. Following (Yu et al., 2020; Chen & Zhang, 2020; Du et al., 2021), we set modality missing arising during the training phase to show that a large amount of unimodal data can assist the training of our multimodal learning framework. In the inference phase, we use Equation (4) to predict the class label of the given test data.

Finally, we run each experiment five times and report average test accuracies to evaluate the performance of our approach and baseline methods. Adam optimizer (Kingma & Ba, 2015) is used to train neural networks with the learning rate of 0.0001. Both the size of modality-complete batch and the size of modality-missing batch are set to 90. The number of epochs is set to 100. All experiments are implemented by Pytorch (Paszke et al., 2019) on a NVIDIA TITAN V GPU card.

## 3.3 BASELINE METHODS

To show the effectiveness of our method, we compare our approach with the following methods which can also handle modality missing to some extent.

- Discarding Modality-incomplete Data (Lower Bound): One simple strategy to handle modality missing is to directly discard the modality-incomplete data, and then only use the modality-complete data for the classification task. This method does not use the information of the data with missing modalities. In our maximum likelihood estimation model, this is equivalent to calculating $Q_{Z|XY}$ without calculating $Q_{Z|X}$. Therefore, this method can also be used as the ablation study of our approach.

- Hirschfeld-Gebelein-Renyi Maximal Correlation (Hirschfeld, 1935; Gebelein, 1941; Rényi, 1959) (HGR MC): HGR MC is a statistical measure which calculates the dependence between different random variables. It has been successfully used for multimodal learning (Ma et al., 2021a; 2020; Wang et al., 2019; Xu & Huang, 2020). Here, we use it further to deal with modality missing. For the modality-complete data, we learn the maximal correlation between $x$, $y$, and $z$. For the modality-missing data, we learn the maximal correlation between $x$ and $z$.

- Zero Padding (ZP): Padding the feature representation of the missing modality with zero is another widely used way to cope with incomplete modalities (Jo et al., 2019; Chen et al., 2020; Shen et al., 2020). For ZP, we consider two forms of $\phi$ to fuse features $f$ and $g$: addition and concatenation. The reason why the form of outer product is not studied here is that if the feature of one modality is zero, the outer product of it and the non-zero feature of another modality is also zero, which makes the modality-missing data useless.

- Autoencoder (AE): An autoencoder is a neural network framework used to learn the representation from the training data. Some previous approaches apply autoencoders to complement

Table 2: The classification performance with missing modality on the RAVDESS dataset.

| Method | | Visual Missing | | | Audio Missing | | |
|---|---|---|---|---|---|---|---|
| | | 95% | 90% | 80% | 95% | 90% | 80% |
| Lower Bound | Addition | 43.01 | 48.79 | 62.66 | 37.11 | 50.40 | 67.63 |
| | Concatenation | 41.27 | 49.94 | 62.66 | 35.61 | 49.48 | 67.63 |
| | Outer product | 37.34 | 45.90 | 63.35 | 36.87 | 47.86 | 68.32 |
| AE | Addition | 52.60 | 58.03 | 67.75 | 87.39 | 88.44 | 90.17 |
| | Concatenation | 53.29 | 58.03 | 67.75 | 87.39 | 88.21 | 90.29 |
| | Outer product | 53.53 | 57.34 | 66.24 | 87.86 | 89.25 | 90.52 |
| HGR MC | Addition | 38.49 | 48.78 | 64.51 | 42.43 | 53.87 | 73.75 |
| | Concatenation | 42.08 | 48.90 | 65.20 | 43.70 | 52.60 | 73.17 |
| | Outer product | 37.34 | 46.59 | 62.43 | 43.01 | 51.33 | 70.98 |
| ZP | Addition | 44.74 | 52.72 | 67.40 | 83.81 | 86.24 | 90.17 |
| | Concatenation | 43.70 | 52.02 | 65.89 | 82.31 | 86.47 | 90.63 |
| Ours | Addition | 56.64 | 60.34 | **70.06** | 89.72 | 90.52 | 91.33 |
| | Concatenation | **58.84** | **61.27** | 69.83 | 89.25 | 89.94 | **91.91** |
| | Outer product | 54.45 | 58.26 | 67.51 | **90.06** | **91.21** | 91.68 |

the missing modality (Liu et al., 2021; Jaques et al., 2017; Pereira et al., 2020; Shi et al., 2019). Following these works, for the modality-complete data, we use modality $x$ as the input of the autoencoder to reconstruct modality $y$. Then we use the trained autoencoder to predict the modality $y$ to impute the modality-missing data. Then we use the imputed data to perform the classification task. It is noted that the AE method has several stages while our method is end-to-end.

For a fair comparison, we make that each method has the same network architecture and training strategy, and report the classification results of each method after the same number of repeated experiments.

## 3.4 EXPERIMENTAL RESULTS

We first conduct classification experiments on the eNTERFACE'05 and RAVDESS datasets by comparing our framework with other methods. The experimental setting is shown in Section 3.2. We report the classification accuracy of each method in each setting. The results are shown in Table 1 and Table 2.

We have the following summarizations from Table 1 and Table 2: (1) The methods of AE, HGR MC, ZP, and ours can improve the classification accuracy compared to the Lower Bound method which only uses the modality-complete data. Our method achieves the highest classification performance among all methods under different settings. The higher the missing rate, the more obvious the gap between other methods and our method. These show that our maximum likelihood estimation approach are more effective to tackle modality missing compared with other methods. (2) Different forms of $\phi$ will affect the classification performance. For example, for our approach, addition and outer product perform better than concatenation on the eNTERFACE'05 dataset. However, on the RAVDESS dataset, the concatenation form of $\phi$ achieves higher classification performance than the addition and outer product forms under some settings. This indicates that in different settings, the discrimination ability of the learned feature representations is different. We need to design the appropriate form of $\phi$ to fuse features of the multimodal data. (3) When the visual modality is missing, the classification accuracy is lower than that when the audio modality is missing, indicating that the visual modality has a more significant contribution to the classification performance, which is consistent with previous works (Zhang et al., 2017; Ma et al., 2020).

In addition, we show the classification confusion matrices using the methods of AE, HGR MC, ZP, and ours when the missing rate of visual modality reaches 95% on the eNTERFACE'05 dataset, as shown in Figure 4. It can be seen that the classification accuracy of each emotion using AE or HGR MC is not high, which indicates that they can only deal with modality missing to a certain extent. The overall classification performance of ZP is lower than ours, but the classification accuracy of

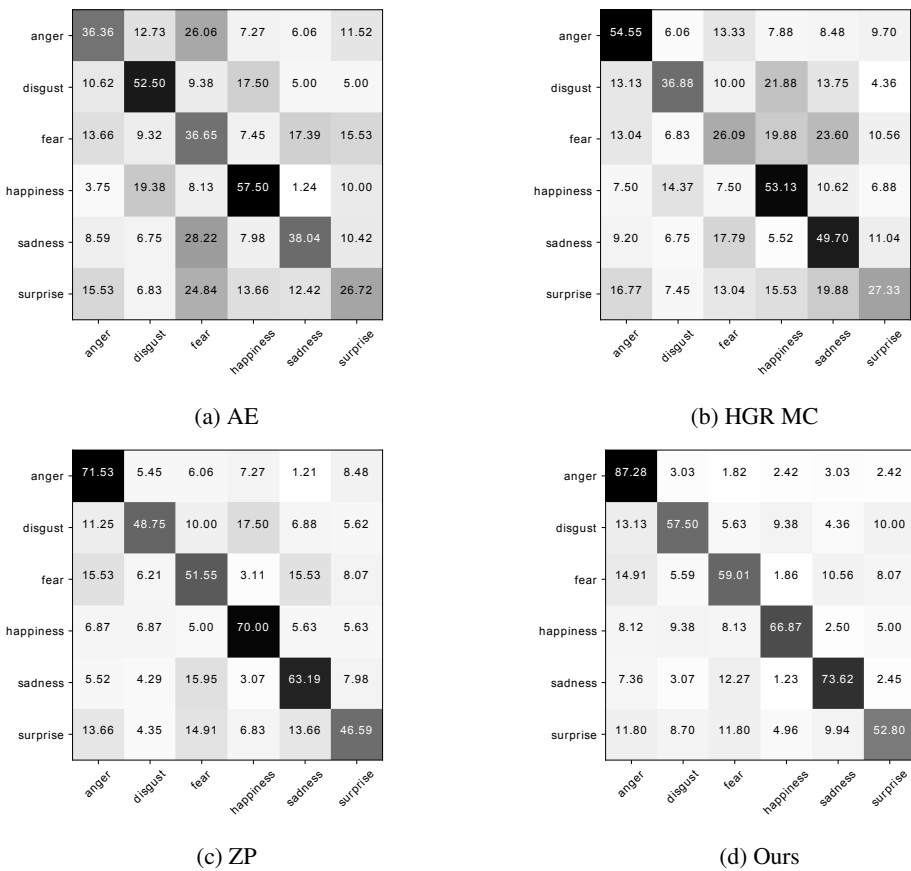

Figure 4: The confusion matrices of different methods on the eNTERFACE'05 dataset.

"happiness" is slightly higher than ours. **This shows that different emotions have different clues for the classification task.**

We then investigate the effect of the backbone in coping with modality missing. In the above experiments, we use ResNet-50 as the backbone of different methods to extract feature representations. Here, we replace ResNet-50 with ResNet-34 (He et al., 2016) and VGG-16 (Simonyan & Zisserman, 2015) respectively, and conduct experiments to compare the performance of different backbones when 95% of the training data has missing visual modality on the RAVDESS dataset, as shown in Figure 5. We can observe that compared with VGG-16 and ResNet-34, ResNet-50 achieves the highest performance. In addition, no matter what kind of backbone is based on, the classification accuracy using our method is the highest, followed by using AE, ZP and HGR MC, and the lowest using Lower Bound, which shows that our approach can take effect for different backbones.

## 4 RELATED WORKS

Multimodal learning has achieved great successes in many applications. An important topic in this field is multimodal representations (Baltrušaitis et al., 2018; Zhu et al., 2020), which learn feature representations from the multimodal data by using the information of different modalities. How to learn good representations is investigated in (Ngiam et al., 2011; Wu et al., 2014; Pan et al., 2016; Xu et al., 2015). Another important topic is multimodal fusion (Atrey et al., 2010; Poria et al., 2017), which combines the information from different modalities to make predictions. Feature-based fusion is one of the most common types of multimodal fusion. It concatenates the feature representations extracted from different modalities. This fusion approach is adopted by previous works (Tzirakis et al., 2017; Zhang et al., 2017; Castellano et al., 2008; Zhang et al., 2016). Modality missing is a key challenge in applying multimodal learning to the real world.

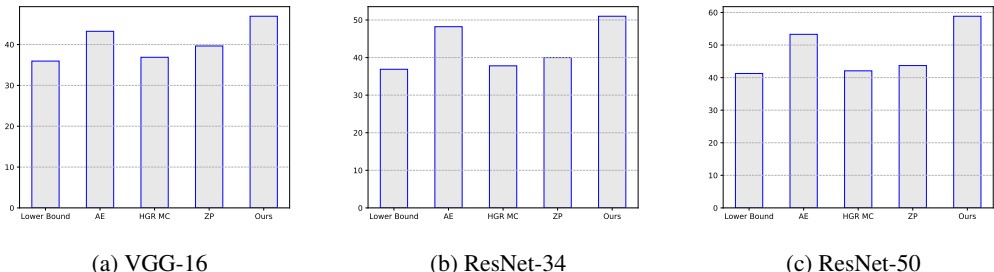

| (a) VGG-16 | (b) ResNet-34 | (c) ResNet-50 |

Figure 5: The performance comparison of different backbones on the RAVDESS dataset.

To cope with the problem of modality missing, a few methods have been proposed. For example, Ma et al. (2021b) propose a Bayesian meta learning framework to perturb the latent feature space so that embeddings of single modality can approximate embeddings of full modality. Tran et al. (2017) propose a cascaded residual autoencoder for imputation with missing modalities, which is composed of a set of stacked residual autoencoders that iteratively model the residuals. Chen & Zhang (2020) propose a heterogeneous graph-based multimodal fusion approach to enable multimodal fusion of incomplete data within a heterogeneous graph structure. Liu et al. (2021) propose an autoencoder framework to complement the missing data in the kernel space while taking into account the structural information of data and the inherent association between multiple views.

The above approaches can combine the information of the modality-missing data to some extent. Our work is significantly different from them. The reason lies in the following two facts. Firstly, by exploiting the likelihood function to learn the conditional distributions of the modality-complete data and the modality-missing data, our method has a theoretical guarantee, which is skipped by previous works. Secondly, the training process of our approach is in an end-to-end manner, while the training processes of most above methods are relatively cumbersome.

## 5 CONCLUSION

Multimodal learning is a hot topic in the academic and industry communities, of which a key challenge is modality missing. In practice, the multimodal data may not be complete due to various reasons. Most previous works cannot effectively utilize the modality-missing data for the learning task. To address this problem, we propose an efficient approach to leverage the knowledge in the modality-missing data during the training stage. Specifically, we present a framework based on maximum likelihood estimation to characterize the conditional distributions of the modality-complete data and the modality-missing data, which has a theoretical guarantee. Furthermore, we develop a generalized form of the softmax function to effectively implement our maximum likelihood estimation framework in an end-to-end way. We conduct experiments on the eNTERFACE'05 dataset and the RAVDESS dataset for multimodal learning to demonstrate the effectiveness of our approach. In the future, we can further extend our framework to other multimodal learning domains.

## REPRODUCIBILITY STATEMENT

We provide our code in "supplement.zip". In this folder, "eNTERFACE_preprocess.py" and "RAVDESS_preprocess.py" extract segment samples from the original videos of the eNTER-FACE'05 dataset and the RAVDESS dataset, respectively. "mle.py" shows the function to compute our maximum likelihood estimation algorithm.

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
