# OpenReview forum: "Maximum Likelihood Estimation for Multimodal Learning with Missing Modality"
_ICLR.cc/2022/Conference — ICLR 2022 Submitted_

### Official Review · Reviewer_APG5 · 2021-10-27

**Correctness:** 3
**Technical Novelty And Significance:** 3
**Empirical Novelty And Significance:** 2
**Recommendation:** 3
**Confidence:** 4

**Details Of Ethics Concerns:**

I do not have any Ethics Concerns as publicly available datasets were used.

**Main Review:**

Strengths:
-	Multimodal learning has achieved great success for many applications and having missing modality is an important challenge to be tackled. This paper presents a simple, end-to-end method, in a way novel and contributing to the field as being based on maximum likelihood, which is not presented by prior art.

Weakness:
-	The biggest limitation of this work is considering only the multimodal data having two modalities as well as not even discussing how the proposed method behaves in case of having more modalities.
-	Although the paper presents a task-free method, the experimental analysis were limited to two datasets, both are addressing the same task: emotion recognition. As also mentioned in the introduction there are several other tasks that the proposed method could have been tested on. Indeed, the related work (such as Ma et al. SMIL: "Multimodal Learning with Severely Missing Modality".) was tested on several different tasks. I suggest authors to either change the paper including the title, abstract and related work and target categorical emotion recognition or comprehensively extend the experimental analysis such that the proposed method would be tested and validated on several other tasks.
-	It is also important to mention that the proposed method was tested only for categorical emotion recognition, while emotion datasets are typically multi-labeled (as humans cannot elicit only one emotion at a time) and also include continuous values therefore, regression task might be targeted too
-	I also found the used datasets limited in terms of their size. In case authors would like to keep the emotion recognition task as the testbed, I suggest them using a much larger dataset called: CMU-MOSEI, also having other modalities than video and audio. Indeed some related work was tested on CMU-MOSEI and/or CMU-MOSI such as Ma et al. SMIL: "Multimodal Learning with Severely Missing Modality".
-	Another limitation regarding the experimental analysis performed is that: as modality only visual and audio data were used. Testing on combinations of several other modalities: text, depth data, data of mocap, accelerometer, gyro-meter would improve the validity of the proposed method.
-	“In addition, the generalized softmax function we propose….” Generalized softmax function might be misleading, it more sounds like the used softmax has tolerance to the diversity of samples belonging to different classes or somehow a domain adaption is being applied. But these are not the cases.
-	It is unclear why authors think that the multimodal softmax is a contribution. Eq. 3 and following equations look like standard softmax was written for multimodal data instead of first fusing the data and representing the fused data as a single feature vector. On the other hand, the fusion of the data is performed through standard strategies: addition, concatenation and multiplication. I expect authors to clarify the contribution in this respect.
-	There are also lack of information regarding how the data is being processed. In detail:
a)	What does “we take central frame” as visual modality mean? Do you take a bunch of frames and use only the central frame? If so what is the motivation behind this? What is the window size? In fact, it is more frequent to apply spatio-temporal processing, for example, processing motion and appearance in facial images for emotion recognition. Thus, I do not understand the rationale behind discarding the temporal information.
b)	Another issue is reading the audio data; it is not clear what the audio data chunk selected to calculate the log Mel-spectrogram.
c)	“On each processed dataset, we split all data into three parts: training set, validation set, and test set. Their proportions are 70%, 15%, and 15%.” Are you randomly picking these splits and applying sort of a k-fold cross validation or these splits are obtained only once and fixed? Do you guarantee that you use exactly the same split for all baseline methods, this is a matter because the used datasets are relatively small? I am aware that prior art on emotion recognition uses 5 or 10 fold cross validation for the same datasets, and I am not sure why authors have selected a different data splitting strategy.
-	  The proposed method was compared with some relatively simpler baselines such as zero padding, but the comparative study should include the SOTA methods, e.g., Ma et al (2021b), Tran et al. (2017), Chen & Zhang (2020), Liu et al. (2021), Suo et al., (2019). Given this lack of comparison, I believe that the claim of authors “……which lead to the information of the modality-missing data not being well exploited to” was not justified as well.
-	I believe authors should include a better discussion why they tackle with the missing data only in training but never take into account that there could be missing modality in testing as well. I think in a practical scenario it is more possible to train a model with a full set of modalities, while during test some of the modalities are either completely or only for some test samples missing.
-	Tables should include the results of unimodal data processing to allow reader to understand which modality perform better than other when used alone, and include the results of processing complete data (i.e., no missing modality) as the upper bound.
-	“When the visual modality is missing, the classification accuracy is lower than that when the audio modality is missing, indicating that the visual modality has a more significant contribution to the classification performance, which is consistent with previous works (Zhang et al., 2017; Ma et al., 2020).” I believe the citations in this sentence is a bit irrelevant. In detail, the authors are not using neither the same feature sets nor the same datasets with the cited works.




**Summary Of The Paper:**

This paper deals with multimodal learning with missing modality in training. Specifically, the proposed method is based on maximum likelihood estimation to obtain the conditional distributions of the so-called "the modality complete data" and "the modality-missing data" in which a multimodal softmax function is defined to implement this framework in an end-to-end manner.

**Summary Of The Review:**

The experimental analysis is limited in terms of several aspects: a) applications: tested only for emotion recognition, b) fixed type of modalities: only audio and video, c) no comparisons with the SOTA

---

### Official Review · Reviewer_wkEE · 2021-11-02

**Correctness:** 3
**Technical Novelty And Significance:** 2
**Empirical Novelty And Significance:** 2
**Recommendation:** 3
**Confidence:** 4

**Main Review:**

1. In Introduction , the authors states that "Compared with unimodal learning, multimodal learning can effectively utilize the multimodal data to achieve better performance.", actually in some cases multimodal data must be utilized properly to make multimodal learning more effective  than unimodal learning. For e.g., researchers have found that the best unimodal model can outperform its multimodal counterpart in this paper: "W. Wang, et al. What Makes Training Multi-modal Classification Networks Hard?" The authors should try to make the statements more accurate.

2. The author have mentioned in Page 4 that "In the following, we will show that we use empirical distribution to implement these underlying marginal distribution in our algorithm.", but the reviewer could not find any descriptions in the following paragraphs.

3. In experiments on eNTERFACES’05, the condition with 100% missing rate should be considered, which could be helpful to demonstrate whether the left 5% data in 95%-missing case is indeed used for the task or just because the other complete modality of data.

4. The authors mentioned several different methods dealing with missing modality in related works, but no experiments to compare the performances between the proposed methods and the mentioned framework. At the same time, the comparative methods in this submission are less persuasive.

**Summary Of The Paper:**

This submission proposed a maximum likelihood estimation framework combined with a generalized softmax function to resolve multimodal emotion recognition with missing modality. Two emotion recognition datasets are used in experiments to make comparison with several baseline methods. The results suggest that the proposed approach outperforms these compared methods. Moreover, according to the authors, the end-to-end nature of this framework makes it more efficient than previous works.

**Summary Of The Review:**

There is a mismatch between the title and the content, given that the proposed methods are only verified in one application scenario, i.e., emotion recognition. However, the multimodal learning is such a big topic, including but not limited to emotion recognition, action recognition, etc. The authors may consider to extend the methods in order to match the title or change the title to a specific area.

Besides, the experiments the authors have conducted is far from extensive, and the comparative methods are not sufficent to support the conclusion.

---

### Official Review · Reviewer_cK5q · 2021-11-03

**Correctness:** 2
**Technical Novelty And Significance:** 2
**Empirical Novelty And Significance:** 2
**Recommendation:** 3
**Confidence:** 4

**Main Review:**

The main contribution of the paper is the proposal of the generalized softmax function, to model the joint distribution of all modalities and the label. The generalized softmax function consists of the product of the marginal distributions of the modalities and the label, as if they were independent, subsequently compensating for this (the dependence among the modalities and the label) via an exponential function, enhanced with feature extraction models. This joint distribution leads to computationally efficient conditional distributions.

However, there are a few concerns about the approaches and the evaluations in this paper:

* The most significant concern is about the baseline comparison.  The authors set these baselines as instances of specifically defined simpler models (or their own model) in order to highlight specific manner in which those models deal with the missing modality. However, there are many prior works focused on solving the exact missing modality problem (see below). The authors should, thus, compare against those baselines instead of deriving their own baseline model instances.  [1] Multimodal Generative Models for Scalable Weakly-Supervised Learning (Wu and Goodman); [2] Private-Shared Disentangled Multimodal VAE for Learning of Latent
Representations (Lee and Pavlovic); [3] MHVAE: a Human-Inspired Deep Hierarchical Generative Model for Multimodal Representation Learning (Vasco, Melo, and Paiva).

* In the experiments (Tab.1 and Tab.2), the visual missing rate and the audio missing rates are likely those in the training set. It is not clear what are the missing rates are for the testing set, if any.  The authors should clarify this.

* In Fig.4 (c) and (d), the classification accuracy of the “happiness” emotion by ZP is higher than that by the proposed method. This single value may be caused by several factors, hidden or accidental. Thus, it may not be sound to claim that “the proposed method is more efficient to exploit the information in most categories”. To support that claim, more investigation is needed and the authors should present it.


**Summary Of The Paper:**

The authors propose a probabilistic framework to improve the classification accuracy in instances when there exists missing data in the multi-modality datasets (where one of the modalities is the predictive label; however, this label is not assumed missing). To this end, they propose a generalized softmax function as the joint distribution of all modalities and the label, from which conditional distributions are derived, for computing the maximum likelihood estimate (MLE). Experimental results on eNTERFACE and RAVDESS datasets demonstrate improvements in classification accuracy over baselines. In addition, the authors investigate the influence of the influence of the backbone models and the fusion functions.


**Summary Of The Review:**

I find the claims in the paper largely unsubstantiated due to the lack of comparison to baselines as mentioned in the Main Review above.

---

### Decision · Program_Chairs · 2022-01-20

**Decision:**

Reject

**Comment:**

Three experts reviewed this paper and all recommended rejection. The rebuttal did not change the reviewers' recommendations. The reviewers was not excited by the proposed probabilistic framework and raised many concerns regarding the comparison with baselines and competing methods, limited size of datasets, and limited scope of one dataset for one task. Considering the reviewers' concerns, we regret that the paper cannot be recommended for acceptance at this time.  The authors are encouraged to consider the reviewers' comments when revising the paper for submission elsewhere.